# The role of ducks in detecting Highly Pathogenic Avian Influenza in small-scale backyard poultry farms

Steven Xingyu Wu[1,2]*, Christopher N. Davis[1,2,3], Mark Arnold[4], Michael J. Tildesley[1,2,3]

**1** The Zeeman Institute, University of Warwick, Coventry, United Kingdom, **2** Mathematics Institute, University of Warwick, Coventry, United Kingdom, **3** School of Life Sciences, University of Warwick, Coventry, United Kingdom, **4** Animal and Plant Health Agency, Addlestone, United Kingdom

\* steve.wu@warwick.ac.uk

## Abstract

Previous research efforts on highly pathogenic H5N1 avian influenza (HPAI) suggest that different avian species exhibit a varied severity of clinical signs after infection. Waterfowl, such as ducks or geese, can be asymptomatic and act as silent carriers of H5N1, making detection harder and increasing the risk of further transmission, potentially leading to significant economic losses. For backyard hobby farmers, passive reporting is a common HPAI detection strategy. We aim to develop a computational, mechanistic model to quantify the effectiveness of this strategy by simulating the spread of H5N1 in a mixed-species, small-population backyard flock. Quantities such as detection time and undetected burden of infection in various scenarios are compared. Our results indicate that the presence of ducks can lead to a higher risk of an outbreak and a higher burden of infection. If most ducks within a flock are resistant to H5N1, detection can be significantly delayed. We find that within-flock infection dynamics can heavily depend on the species composition in backyard farms. Ducks, in particular, can pose a higher risk of transmission within a flock or between flocks. Our findings can help inform surveillance and intervention strategies at the flock and local levels.

## Author summary

We addressed the gap in our understanding of within-flock transmission dynamics of H5N1, particularly for small-scale, backyard farms, where it is reasonably realistic for multiple species of birds to be housed together. These smaller flocks may differ from their larger, industrialised counterparts in their structure and management, and may play a key role in the persistence and spread of H5N1. Notably, we know from the literature that waterfowl, such as ducks, can be asymptomatic after contracting H5N1 and thus act as 'silent carriers' of the virus, which could amplify

**Data availability statement:** The code to generate all simulated data can be found in https://github.com/stevewu2001/Within-flock-simulation.

**Funding:** SXW was funded by the Engineering and Physical Sciences Research Council (EPSRC) through the Mathematics of Systems Centre for Doctoral Training (MathSys CDT) [grant number EP/S022244/1] (website: https://www.ukri.org/councils/epsrc/). CND and MJT were supported by a joint Ecology and Evolution of Infectious Diseases (EEID) award from the U.S. National Science Foundation (NSF) and the Biotechnology and Biological Sciences Research Council (BBSRC) [grant number BB/X005224/1] and a BBSRC grant [BB/X016137/1]. (websites: https://www.nsf.gov/ and https://www.ukri.org/councils/bbsrc/). The funders had no role in study design, data collection and analysis, decision to publish, or preparation of the manuscript.

**Competing interests:** The authors have declared that no competing interests exist.

the risk posed to other species of birds, mammals, and humans. We used a stochastic mechanistic model that accounted for such possibilities and simulated the possible outcomes of an outbreak. We found that while the presence of chickens is more likely to lead to high mortality upon infection, ducks can make H5N1 harder to detect within a flock, and thus cause a greater burden of infection, which increases the risk of potential between-flock and between-site transmissions. Our findings are consistent with the current literature and can help inform surveillance and control strategies.

## Introduction

Highly Pathogenic Avian Influenza (HPAI), the H5N1 subtype, remains a threat to global health security as of 2025. The current H5N1 strain (A/goose/Guangdong/1/96) was first isolated from a domestic goose in China in 1996. Since then, numerous large outbreaks occurred in various countries in Asia in 2003, and subsequently led to cases found in Africa, Europe, and the Americas in the following decades, including nationwide outbreaks in many countries [1,2]. Although infections are most commonly found among birds, mammals and humans can also be affected. For example, a notable incident occurred in Hong Kong in 1997, where six people died after contracting an infection [2]. In 2024, an outbreak was identified in the US in which dairy cows and two humans contracted H5N1 [3]. Other mammals, including household pets, can also be susceptible to H5N1, as evident in an outbreak in a mink farm in Spain, 2023 [4], and in two cat shelters in South Korea, 2023 [5]. These events are examples of HPAI H5N1 causing significant economic losses and disrupting food security. The rare but significant bird-to-human spillover cases shows its pandemic potential, which is further signified by the recent cases of mammalian infections. Backyard and smallholder flocks are a unique concern in the context of HPAI H5N1 epidemiology, since they are widespread globally, often have limited biosecurity compared to commercialised farms, and potentially have consistent surveillance measures. These factors make them a persistent reservoir of viral circulation and a risk for widespread transmissions and zoonotic exposure. Therefore, understanding the transmission dynamics of the virus is vital in designing effective control and prevention strategies, and minimising potential human exposure to H5N1, especially in backyard flocks.

HPAI H5N1 is known to have a varied effect depending on the species and breed of the infected bird. Many existing studies, such as Bouma et al. [6] and Yu et al. [7], have shown that species such as chickens and turkeys are likely to exhibit noticeable clinical signs and die within days of an infection. In comparison, there exists some evidence that waterfowl such as ducks and geese can be asymptomatic to H5N1, meaning that no clinical signs would be observed in some cases, while still being highly infectious. This means that ducks can potentially act as 'silent carriers' of H5N1, making infections and outbreaks harder to detect promptly [8]. The presence of asymptomatic but infectious birds within a flock may also significantly alter the trajectory of an outbreak. Despite this, relatively few studies have investigated

the epidemiological implications of housing multiple bird species together, particularly in small-scale backyard farms where such an arrangement is reasonably realistic [9]. One of the goals of this study is to investigate the potential risk of 'silent spread' of H5N1 resulting from the presence of ducks in a small, mixed-species flock, and how the infection dynamics changes for flocks with different species population distributions.

Mathematical and computational models have been widely used to study the spread of infectious diseases, including HPAI H5N1, providing insights into key epidemiological parameters such as the transmission rate and infectious period. In the context of HPAI, several studies have focused on fitting mathematical models to mortality data in large commercial flocks [10,11], while others have used experimental data to estimate key parameters such as transmission rate, latency period, and infectious period under controlled conditions [12,13]. A recent review by Kirkeby et al. [14] collected parameter estimates from previous work on HPAI of all subtypes, which were then used for a simulation study [15] to characterise the different epidemiological effects of HPAI subtypes, including H5N1. These works have been instrumental in the design of detection strategies and intervention policies. However, existing current research typically focuses on single-species flocks and overlooks the potentially varied outcomes introduced in mixed-species settings. Furthermore, most current models are geared towards high-density industrialised farms, leaving a gap in our understanding of disease dynamics in small-scale backyard flocks. Between October 2021 and April 2022, approximately a quarter of all HPAI-infected premises in the UK are considered 'smallholding' or 'backyard' [9]. In Southeast Asian countries such as Indonesia, backyard farms were found to have a higher contact rate than other types of flocks and were most at risk for transmissions of HPAI [16], demonstrating the significance of studying these types of flocks. In this study, we aim to address the current gap in literature by focusing on backyard, mixed-species flocks and investigating the effectiveness of passive surveillance. By developing a mechanistic model that allows the possibility of inter-species interactions and infection dynamics in mixed-species flocks, we explore how different population compositions of species influence outbreak probability, detection time, and infection burden.

In this study, we develop a stochastic, mechanistic model to simulate the within-flock spread of HPAI H5N1 in small, mixed-species populations. By varying the proportion of species such as chickens and ducks, we aim to evaluate how species composition influences: (i) the overall disease dynamics assuming no intervention, (ii) the time to HPAI detection via passive surveillance based on mortality signals, and (iii) the total burden of infection, measured by the total time birds spent infectious, up to the point of detection. The potential output of the model can inform the effectiveness of passive surveillance as an HPAI detection strategy, and is of practical relevance for designing early warning systems, optimising surveillance strategies, and implementing intervention policies such as vaccinations or culling. Ultimately, our goal is to improve our understanding of how heterogeneity in species susceptibility, clinical signs, and mortality rate can influence the trajectory and detectability of HPAI outbreaks in non-industrial poultry settings.

## Methods

### Model construction

We considered a backyard farm that housed either chickens, ducks, or a mixture of both. For the purpose of simulations, we used a fixed number of 40 total birds to simulate a smallholder backyard premises, where the birds can be either layers or broilers. A Susceptible–Exposed–Infectious–Recovered–Dead (SEIRD) model was used as a baseline with added possibilities for a bird to be asymptomatic while infectious. Therefore, individual birds were divided into the following compartments: susceptible ($S$), exposed ($E$), infectious and symptomatic ($I$), infectious and asymptomatic ($I^*$), recovered ($R$), and dead ($D$). A subscript was used to denote the species of the bird: $c$ for chickens and $d$ for ducks (e.g. $S_c(t)$ denotes the number of susceptible chickens at time $t$). An exposed bird would become infectious at the rate $\sigma_s$, with the probability of being symptomatic $p_s$, for $s = c, d$, where the species was chickens or ducks, respectively. An infectious bird would recover or die with the rate $\gamma_s, \gamma_s^*$, and the probability of death $\delta_s, \delta_s^*$, based on the bird species and the existence of clinical signs. The model assumed a homogeneously mixing bird population with a density-dependent contact rate, which

is more appropriate for small populations on backyard premises. Therefore, as justified in Fournie et al. [17], the force of infection at time $t \geq 0$ on species $x \in \{c, d\}$ was given by:

$$\lambda_x(t) = \frac{1}{N-1}(\beta_{cx}I_c(t) + \beta_{cx}^*I_c^*(t) + \beta_{dx}I_d(t) + \beta_{dx}^*I_d^*(t))$$

where $N = 40$ was the initial size of the flock (The $-1$ was required because a bird cannot 'contact' itself), and $\beta_{yx}, \beta_{yx}^*$ were the transmission rate from symptomatic and asymptomatic birds of species $y$ to species $x$, respectively. We assumed all other inter-compartment transitions occur with an exponentially distributed waiting time. Thus, we had the transition rules for chickens:

$$S_c \xrightarrow{\lambda_c(t)} E_c$$
$$E_c \xrightarrow{\sigma_c} I_c$$
$$I_c \xrightarrow{\gamma_c} D_c$$

and for ducks:

$$S_d \xrightarrow{\lambda_d(t)} E_d$$
$$E_d \xrightarrow{p_d\sigma_d} I_d$$
$$E_d \xrightarrow{(1-p_d)\sigma_d} I_d^*$$
$$I_d \xrightarrow{\delta_d\gamma_d} R_d$$
$$I_d \xrightarrow{(1-\delta_d)\gamma_d} D_d$$
$$I_d^* \xrightarrow{\gamma_d^*} R_d$$

The explanations for the rate parameters are found in Table 1, and the compartmental diagram that captures the transition rules is presented in Fig 1. Note that some transition rules are omitted due to the redundancy caused by the choice of parameters, which can be inferred from Table 1.

## Parameter values

Owing to a lack of real-world data on within-flock infection dynamics for backyard farms, we relied on parameter estimations from the literature to inform our parameter choices. The baseline parameters on chicken infection dynamics $(\beta_{cc}, \sigma_c, \gamma_c)$ were chosen based on results by Bouma et al. [6] and Tiensin et al. [10]. Specifically, Bouma et al. [6] presented an experimental study that aimed to provide a parameter estimation of an SEIR model for chickens, and Tiensin et al. [10] used Thailand field data to estimate the transmission rate for different values of the infectious period in an SIR model by using the back-calculation method. The values for $\sigma_c$ and $\gamma_c$ were taken directly from the experimental study. However, the estimated value for $\beta_{cc}$ from the experimental study was based on an inoculated chicken and a susceptible chicken in the same cage, which might not reflect field scenarios. Hence, the value for $\beta_{cc}$ was taken from the field study, using the assumed $\gamma_c^{-1} = 2$, which matched with the experimental study. However, it is worth noting that different breed

**Table 1. Parameter values used in the model, based on literature sources or specified assumptions.**

| Parameter | Description | Value | Source |
|---|---|---|---|
| $\beta_{cc}$ | Chicken-to-chicken transmission rate | 1.15 (1.02–1.30) | [6,10] |
| $\sigma_c^{-1}$ | Chicken latency period | 0.24 (0.099–0.48) | [6] |
| $\gamma_c^{-1}$ | Chicken infectious period | 2.1 (1.8–2.3) | [6] |
| $p_c$ | Chicken symptomatic probability | 1 | Assumed [8] |
| $\delta_c$ | Chicken case-fatality probability | 1 | Assumed [8] |
| $\beta_{dd}$ | Symptomatic duck-to-duck transmission rate | 4.1 (2.8–5.8) | [11] |
| $\sigma_d^{-1}$ | Duck latency period | 0.17 (0.03–0.38) | [11] |
| $\gamma_d^{-1}$ | Symptomatic duck infectious period | 4.3 (2.8–5.7) | [11] |
| $p_d$ | Duck symptomatic probability | 0, 0.2, 0.4, 0.6, 0.8 | Assumed |
| $\delta_d$ | Symptomatic duck case-fatality probability | 0.70 (0.61–0.78) | [11] |
| $\kappa$ | Rescaling for asymptomatic duck parameters | 3 (1-5) | Assumed |
| $\beta_{dd}^*$ | Asymptomatic duck-to-duck transmission rate | $\dfrac{\beta_{dd}}{\kappa}$ | Assumed |
| $\gamma_d^{*-1}$ | Asymptomatic duck infectious period | $\kappa\gamma_d^{-1}$ | Assumed |
| $\delta_d^*$ | Asymptomatic duck case-fatality probability | 0 | Assumed |
| $\beta_{cd}, \beta_{dc}, \beta_{dc}^*$ | Cross-species transmission rates | $\beta_{dd}, \beta_{cc}, \dfrac{\beta_{cc}}{\kappa}$ | Assumed |

Parameter values reflect published estimates where available, with assumptions and derived formulas applied to asymptomatic and cross-species transmission parameters. The rate unit is day$^{-1}$. Sensitivity analysis includes variation in symptomatic probability and rescaling factor $\kappa$.

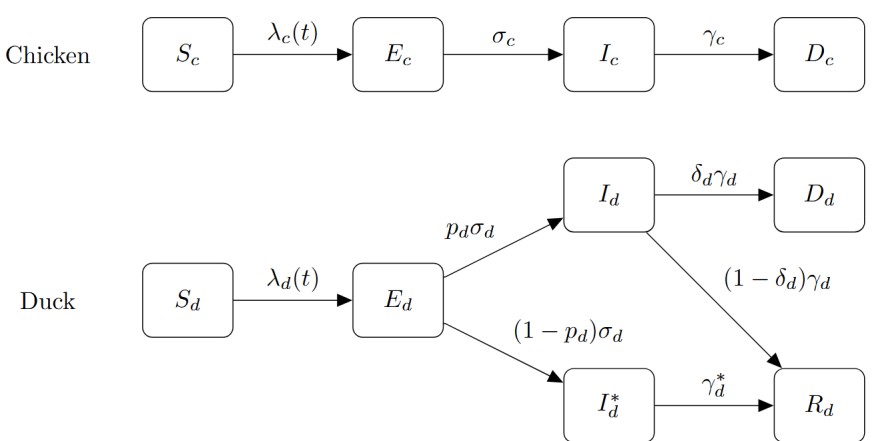

**Fig 1. Model compartmental diagram.** A compartmental diagram of the disease dynamics of a mixed-species flock. The top and bottom parts represent the chicken and duck populations, respectively. We assumed that all infected chickens became symptomatic, while ducks might not exhibit clinical signs with probability $(1 - p_d)$.

of chicken may exhibit different susceptibility to infections [18], so the actual parameter set may slightly differ from our chosen value.

Almost all relevant studies suggested that H5N1 is highly lethal to chickens. Experimental results by Bouma et al. [6], Jeong et al. [19], and Forrest et al. [20], as well as reviews such as the work from Kim et al. [8], implied a 100% mortality rate for infected chickens. While works from Spekreijse et al. [13] and Yu et al. [7] demonstrated that a small proportion of chickens might be asymptomatic depending on the strains of H5N1, but near 100% mortality rate for chickens with clinical signs was still observed. Therefore, we assumed in our model that chickens would always show clinical signs and eventually die of H5N1. For ducks, it was found that the possibility of showing clinical signs and dying from H5N1 varies depending on the breed of the duck [8,21], the strains of the virus [7,21–23], or the route of infection [19]. Therefore, we

conducted a sensitivity analysis on the duck symptomatic probability ($p_d$) to account for the different scenarios, choosing values $p_d = 0, 0.2, 0.4, 0.6$, and $0.8$.

Current literature on parameter estimation of infection dynamics in duck flocks is scarce. One study on HPAI H5N8 from Vergne et al. [11] estimated using field data that the transmission rate was 4.1, the latency period was 0.17, the infectious period was 4.3, and the case-fatality rate was 0.7. Although the HPAI serotype was different, the results from the study were reasonably in agreement with an experimental study from van der Goot et al. [12]. The estimated $R_0$ number for duck flocks was also significantly larger than that of chicken flocks, in agreement with a recent review by Kirkeby et al. [14]. Given the high case-fatality rate assumed in the study, we assumed that the viral strain of focus in the work from Vergne et al. [11] was highly pathogenic to ducks, so the parameter estimations provided would apply specifically to symptomatic ducks.

For asymptomatic ducks, there was a lack of parameter estimation for the transmission rate or infectious period. Based on the work from Hulse-Post et al. [22], we expected that the infectious period for asymptomatic ducks could be as long as 17 days. Therefore, we defined a rescaling factor $\kappa = 3$, where we assumed that the transmission rate $\beta^*_{dd}$ and infectious period $\gamma^*_d$ from asymptomatic ducks were $\frac{\beta_{dd}}{\kappa}, \kappa\gamma^{-1}$ respectively, which ensured the value for $R_0$ was fixed. While $\kappa = 3$ will be assumed for all simulations and results except for the sensitivity analysis, see S5 Fig and S6 Fig from supporting information). Finally, given the homogeneous-mixing population assumption, we set the cross-species transmission rate $\beta_{cd} = \beta_{dd}, \beta_{dc} = \beta_{cc}$, and $\beta^*_{dc} = \frac{\beta_{cc}}{\kappa}$, assuming the same contact rate between any two birds and the probability of infection dependent on the receiving bird.

## Simulations and analysis

The model was simulated using the Gillespie algorithm [24]. Given a set of parameter choices, we set the number of chickens in the population to be 0, 10, 20, 30, and 40, out of 40 total birds, with the remainder being ducks. For the initial condition, one bird was chosen at random to become exposed, and the rest of the flock remained susceptible. The simulation then continued until no disease was present in the system. We performed 5000 realisations of the simulation model and stored them as a time series for each set of parameter choices and species compositions, which can be directly analysed by extracting summary statistics.

Among the 5000 simulations, we considered those that resulted in five or more birds ever being infected as an outbreak. We then filtered out the time series with no outbreaks and considered the interquartile range of the detection time: the day when the farm owner detects a certain number of deaths in the flock. We emulated the behaviours of different types of farm owners based on the level of cautiousness and likelihood of compliance, by considering scenarios where they would report a detection when two, four, or six birds died in the flock. The detection time for each detection rule was then recorded. We also calculated the undetected burden of infection within the flock, defined by the total time birds spent infectious up to the time of detection, measured in bird-days. This was done for every simulation with the death threshold for detection set at two, four, or six, similar to the detection time calculation.

We conducted sensitivity analysis on transmission rates ($\beta_{cc}, \beta_{dd}$), latency ($\sigma_c^{-1}, \sigma_d^{-1}$) and infectious ($\gamma_c^{-1}, \gamma_d^{-1}$) periods, duck case-fatality probability ($\delta_d$), and the rescaling factor for asymptomatic duck parameters ($\kappa$) using the Latin Hypercube Sampling (LHS) method [25], within the ranges specified in Table 1. For each parameter, its range was divided into 50 equal-probability intervals, and one value was sampled from each interval. Samples across parameters were combined via LHS to ensure uniform coverage of the multidimensional parameter space without repetition. Model outputs were then used to quantify parameter sensitivity by computing Spearman rank correlations (PRCC-style) between sampled parameter values and model outcomes: final deaths, detection time, and burden of infection. We tested outcomes for scenarios with $p_d = 0.2$ and chicken population sizes of 0 and 40.

In addition, we examined the effect of flock size on detection time and burden of infection. Flock size was varied from 20 to 200 birds in increments of 20, and for each flock size, simulations were conducted with chicken proportions of 0, 0.25, 0.5, 0.75, and 1, with other parameters held constant and $p_d = 0.2$.

The Python code that achieves the simulations can be found on GitHub: https://github.com/stevewu2001/Within-flock-simulation.

## Results

### Summary statistics

We investigated the final size of the simulations, recording the final death number (Figs 2 and 3 for each simulation. In most cases, most birds in the flock would be infected at some point during a successful outbreak (meaning that they will be either dead or recovered in the end). A higher duck symptomatic probability tends to lead to a larger number of deaths in a flock. This is evident from both the histograms in Figs 2 and 3 and the heat map in Fig 4. Interestingly, Fig 4 suggests that flocks with 30 chickens and 10 ducks result in the highest average number of deaths for all values of duck symptomatic probability.

### Detection time

We found that the outbreak detection time was highly dependent on the choice of parameters in the sensitivity analysis, specifically the probability of a duck showing clinical signs upon infection. Firstly, as per expectations, farmers who are able to detect H5N1 with lower death thresholds had an overall shorter detection time since the initial infection, as shown in Fig 5. We found that the difference between detection thresholds in duck-only flocks is most significant if the duck's symptomatic probability is small (Fig 6 and top plot of Fig 5). We also found that duck-only flocks can result in overall longer detection times, with H5N1 left undetected for 10 days (median), and for as long as 40+ days in the worst-case scenarios. In comparison, the detection time is lower if chickens are present in the flock. Notably, detection is expected within approximately 7 days since the initial infection for chicken-only flocks. However, if the duck symptomatic probability is high, then the overall mortality rate will also increase, and thus the detection time is significantly lowered (bottom plot of Fig 5). The presence of chickens instead increases the detection time slightly, but the overall expected detection time is still fewer than approximately 7 days.

### Burden of infection

In general, due to the long infectious period, the transmissibility of H5N1, and the potentially delayed detection time in duck flocks, their presence in a flock contributes more to the burden of infection compared to chickens. If the duck symptomatic probability is low, the total time birds spend infectious is significantly higher in duck-only flocks, approximately 18 times higher than that of chicken-only flocks (top plot of Fig 7). While the number is reduced in the case of high duck symptomatic probability, a duck-only flock would still have approximately 4 times the burden of infection than chicken-only flocks (bottom plot of Fig 7). Note that we only accounted for time series with detected outbreaks, so if an outbreak occurred but detection failed, then the burden of infection accounting for such cases will be higher in duck-only flocks. As before, the heat map for the burden of infection before detection is shown in Fig 8. The result is consistent with the findings in Fig 7 for other sets of parameter choices. We have chosen to investigate only detected outbreaks, as undetected outbreaks are rare except for the $p_d = 0.2$ case in duck-only flocks, where detection evasion occurs due to the higher likelihood of having asymptomatic ducks. The burden of infection would then be calculated over the entire outbreak. This will not affect the qualitative result, as this case already yields the largest burden of infection with only detected outbreaks, as demonstrated by Fig 8.

 

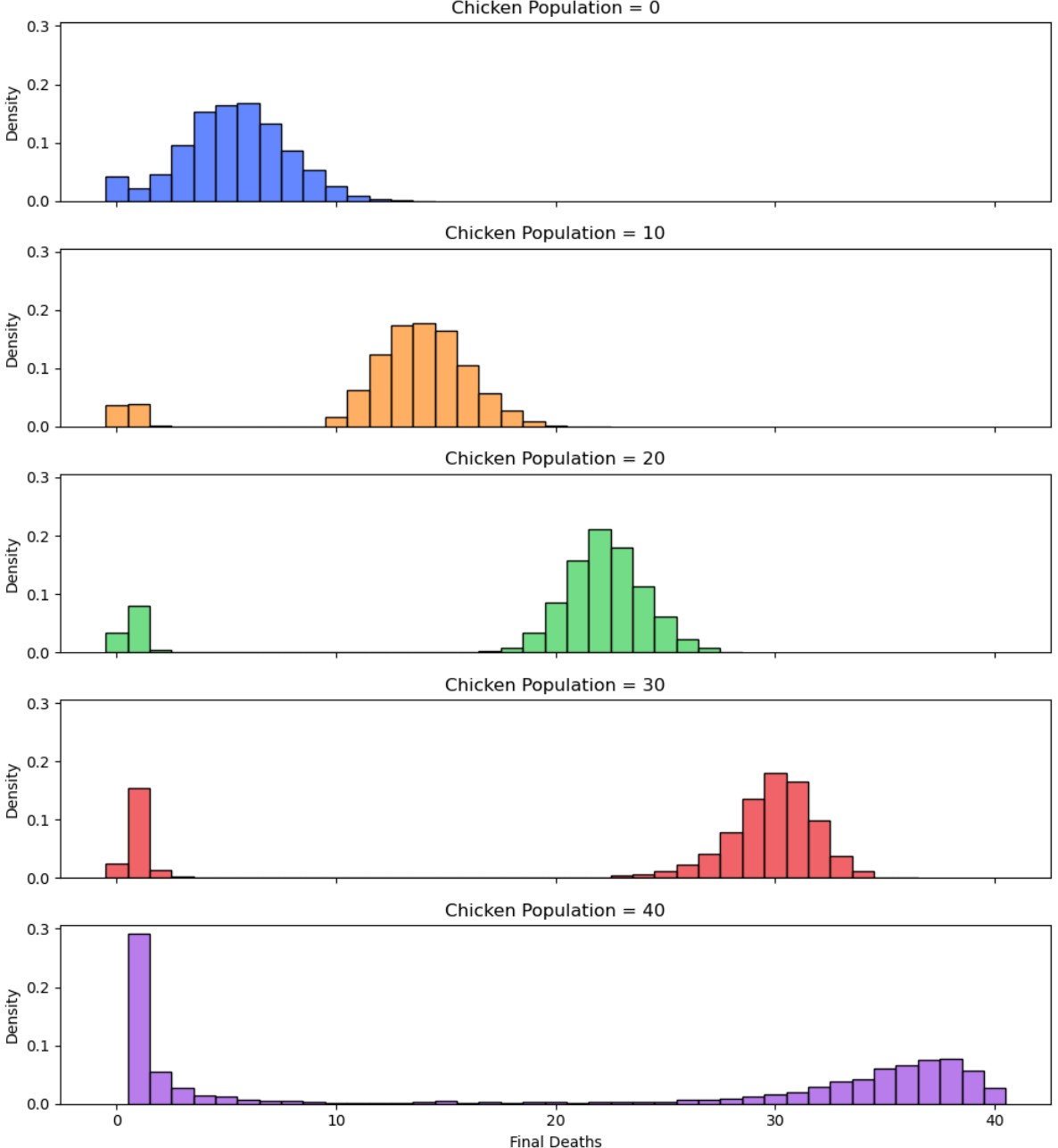

**Fig 2. Histogram final deaths** $p_d = 0.2$. Histogram of distribution of final death numbers with varied number of chickens in flocks of size 40. Duck symptomatic probability set to 0.2.

## Sensitivity analysis

As explained in the Methods section, we briefly explored the impact of the uncertainties of the parameters and total flock size on our results. The Spearman correlations (denoted as $\rho$) between the values of the sampled parameters and key

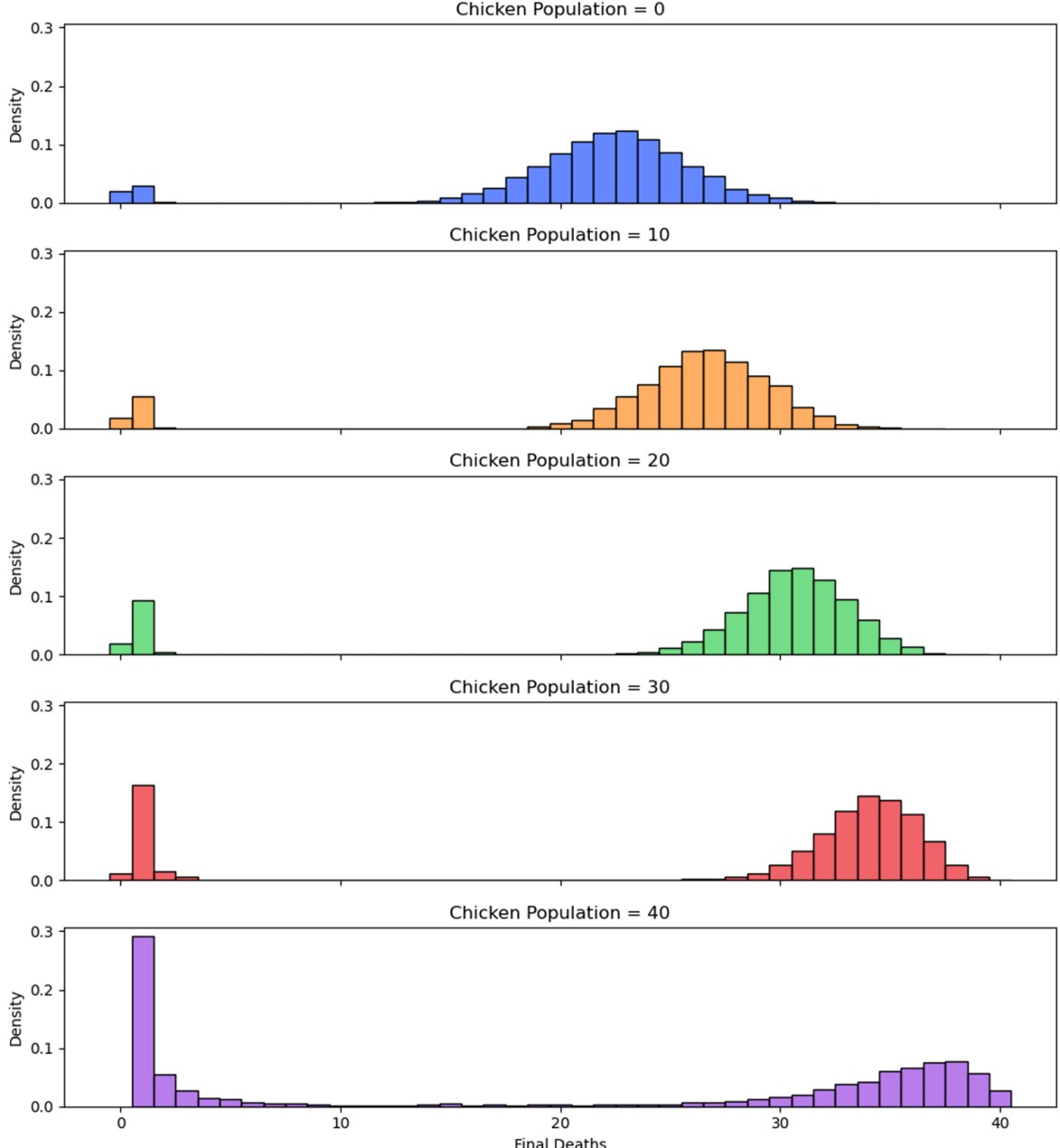

**Fig 3. Histogram final deaths** $p_d = 0.8$. Histogram of distribution of final death numbers with varied number of chickens in flocks of size 40. Duck symptomatic probability set to 0.8.

statistics were shown in Figs 9 and 10. For chicken-only flocks, we found weak correlations for all parameters ($|\rho| < 0.15$). Parameters such as $\kappa$, which are biologically irrelevant in the all-chicken setting, nonetheless showed small correlations ($|\rho| \approx 0.1$) when considering detection times, which we attribute to sampling noise and stochasticity in simulation

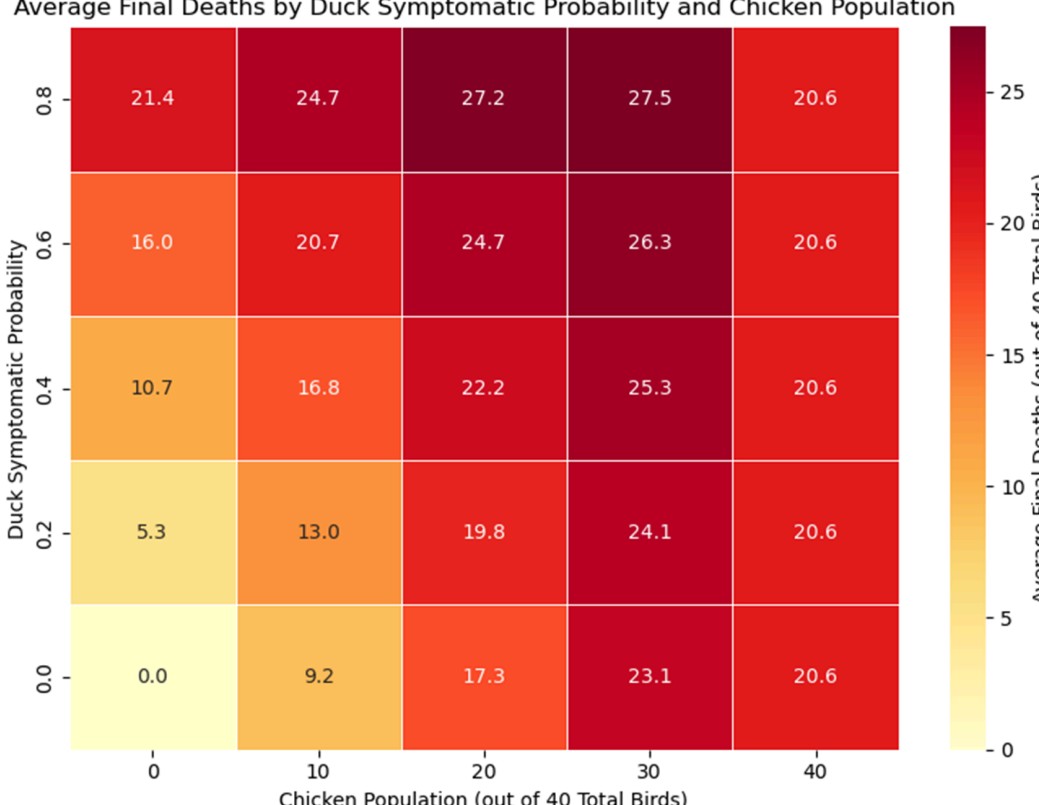

**Fig 4**. **Heat map final deaths.** Heat map showing the average number of deaths in all simulation results given the parameter combinations of chicken populations and duck asymptomatic probability.

outcomes. For duck-only flocks, detection times were found to be moderately sensitive ($|\rho| \approx 0.3$) to the duck-to-duck transmission rate $\beta_{dd}$ and $\kappa$. Likewise, final deaths and burden of infections were mostly influenced by symptomatic duck case-fatality probability $\delta_d$ and infectious period $\gamma_d^{-1}$ respectively, albeit with less significance ($|\rho| \approx 0.15$ and 0.2).

## Discussion

Our study differs from current literature with our unique focus on backyard, mixed-species flocks, the biological differences of each species, and the effectiveness of passive surveillance. To our knowledge, this is the first modelling study to quantify H5N1 transmission and passive surveillance performance in backyard mixed-species flocks, explicitly incorporating species-specific differences in infectiousness, symptoms, and mortality. We showed that ducks, or other similar species, may pose a substantial risk in the context of within-flock infections of HPAI. In particular, duck-only flocks have a higher chance of having an H5N1 outbreak (see S3 Fig and S4 Fig from supporting information), causing delayed detection (Figs 5 and 6) and increasing the risk of undetected outbreaks, and having a significantly higher burden of infection up to the point of detection (Figs 7 and 8), especially if ducks are more likely to be asymptomatic and thus can act as a 'hidden' source of infection. The undetected outbreaks and silent transmissions are signs to focus on duck flocks if alternative surveillance strategies such as active surveillance is utilised. Additionally, having a small population of ducks in a chicken-dominant flock can amplify mortality, as shown in Fig 4, where maximum final deaths are observed for flocks with 30 chickens and 10 ducks given a fixed duck symptomatic probability. The simulation results match expectations,

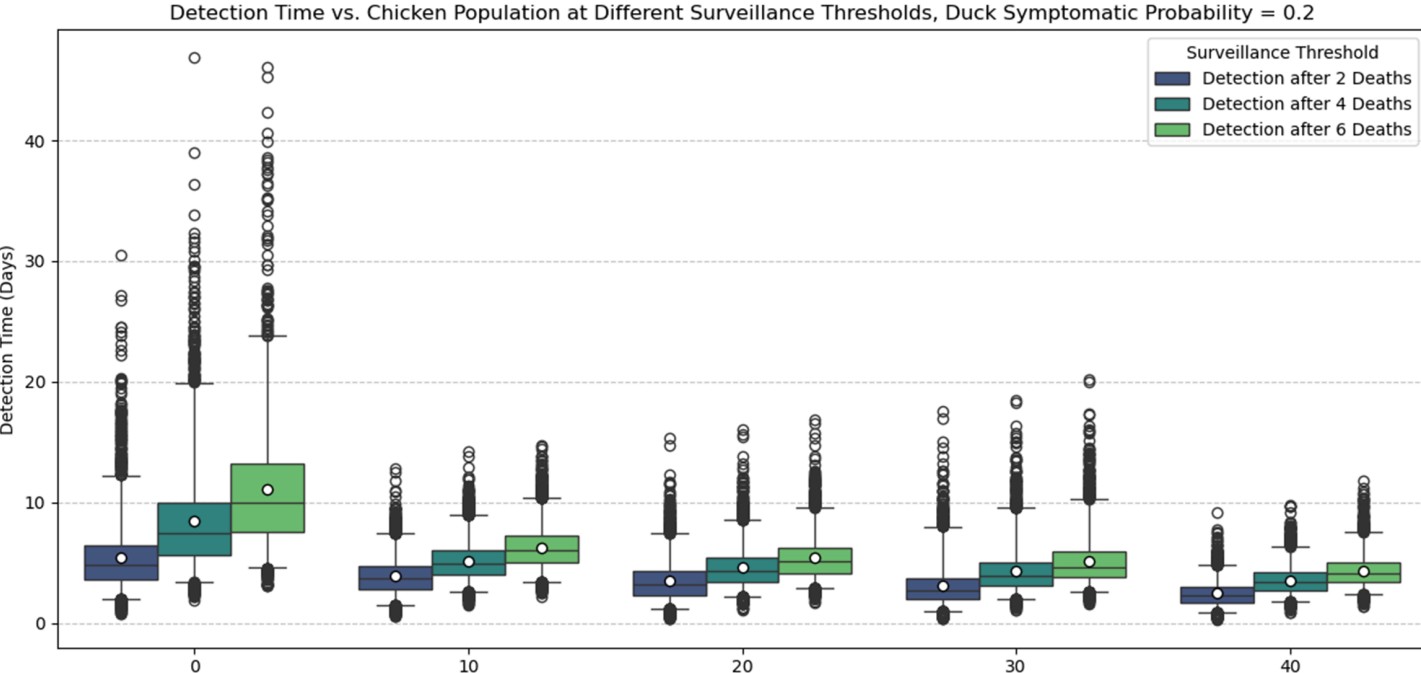

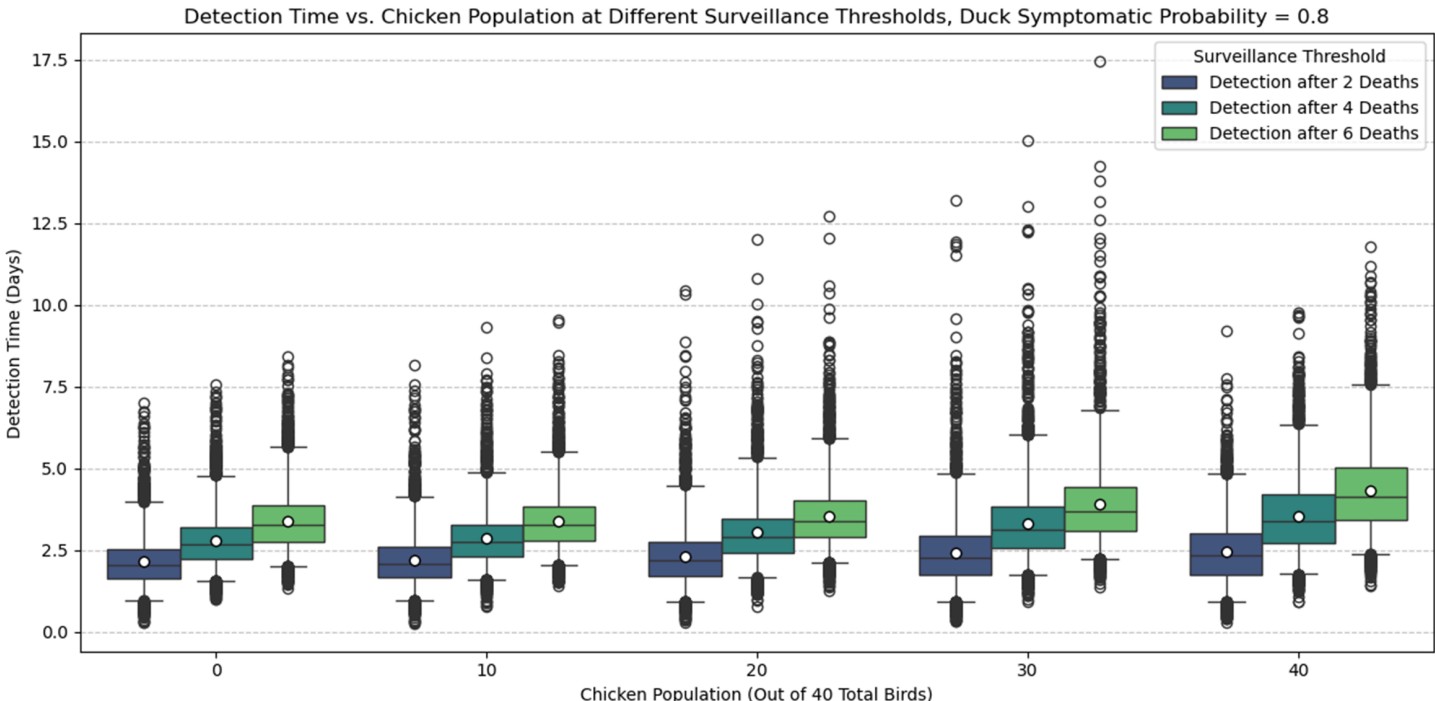

**Fig 5. Box plot detection time.** Detection time for different death thresholds for detection and chicken population out of 40 total birds. Top plot: the duck's symptomatic probability $p_d = 0.2$; Bottom plot: the duck's symptomatic probability $p_d = 0.8$. The boxes indicate the interquartile range (25th and 75th percentiles), and the whiskers indicate the 2.5th and 97.5th percentiles. The white dot in the box indicates the average detection time, which also corresponds to the fourth and first row of Fig 6 (for detection after 2 deaths).

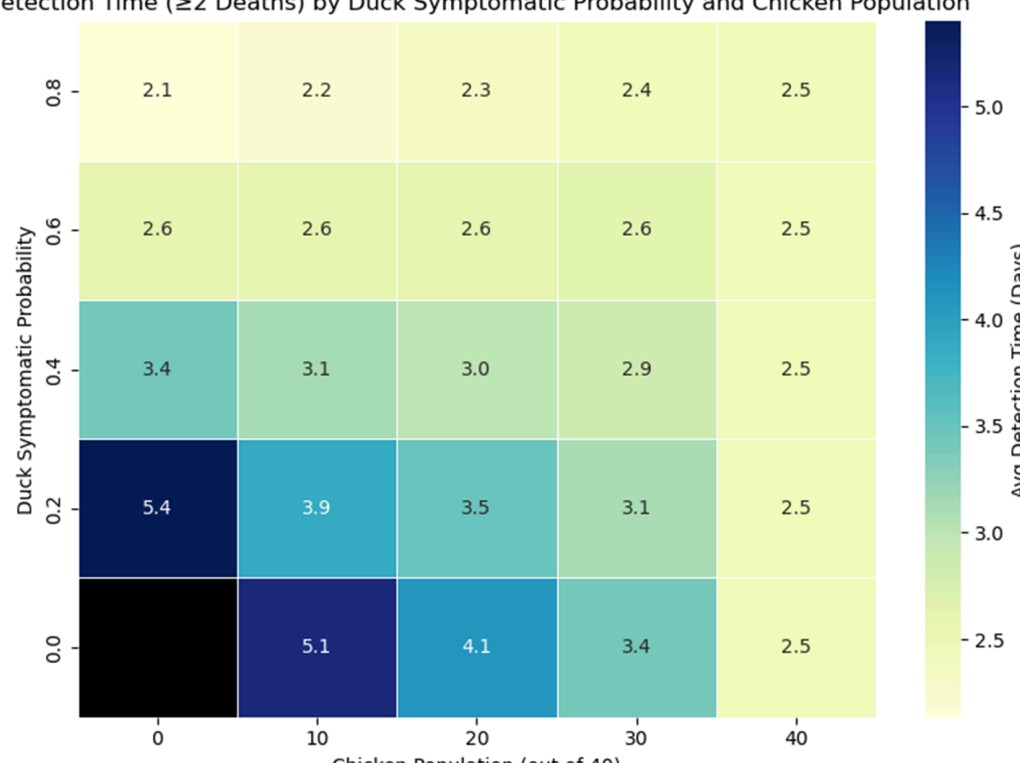

**Fig 6**. **Heat map detection time.** Heat map showing the average detection time (with detection occurring when two deaths have occurred) in all simulation results given the parameter combinations of chicken populations and duck symptomatic probability. Note that no detection would occur for duck flocks with duck symptomatic probability equal to zero (Black grid).

as ducks have a longer infectious period and lower mortality rate. These biological traits directly influence our simulation outcomes, as the prolonged infectious period allows ducks to infect other birds over an extended time frame, while the lower mortality means outbreaks can remain undetected for longer under a passive surveillance system that triggers on deaths. We hypothesise that ducks can act as a more persistent intermediate source of infection, which could lead to more widespread infections when a larger chicken population coexists with the ducks. We also found that ducks tend to contribute to a higher unnoticed burden of infection, again driven by the longer infectious period and lower mortality rate, which explains the higher cumulative infection burden, delayed detection, and increased potential for between-flock transmission and zoonotic exposure through direct or indirect contact between flocks and humans. It is important to note that our analysis focuses on detected outbreaks, which may underestimate the true burden of infection, particularly in duck-only flocks. The undetected outbreaks, while only occurring in duck flocks under certain parameter assumptions, could prolong viral circulation and increase the risk of transmission to nearby flocks. In general, our findings are consistent with many other studies, in which ducks are shown or implied to be a more significant risk factor for HPAI H5N1 transmission [8,11,26].

The most sensitive parameter in our study is $p_d$, the probability that a duck will show clinical signs after infection. Specifically, detection evasion tends to only occur in duck-only flocks if $p_d$ is small. This is likely due to low duck mortality, and most birds in the flock will recover without being detected. For larger values of $p_d$, we found that the detection time becomes shorter in duck-dominant flocks than in chicken-dominant flocks. A possible explanation is that ducks have a higher transmission rate and a longer infectious period, increasing the rate of death among birds. We also saw a lower

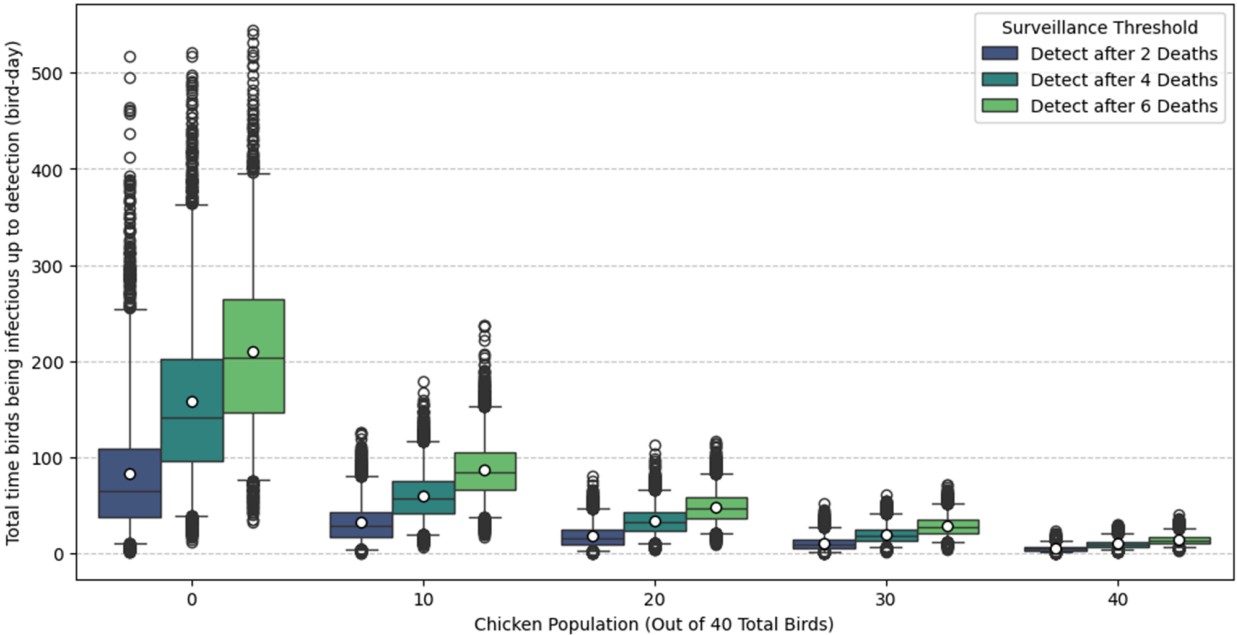

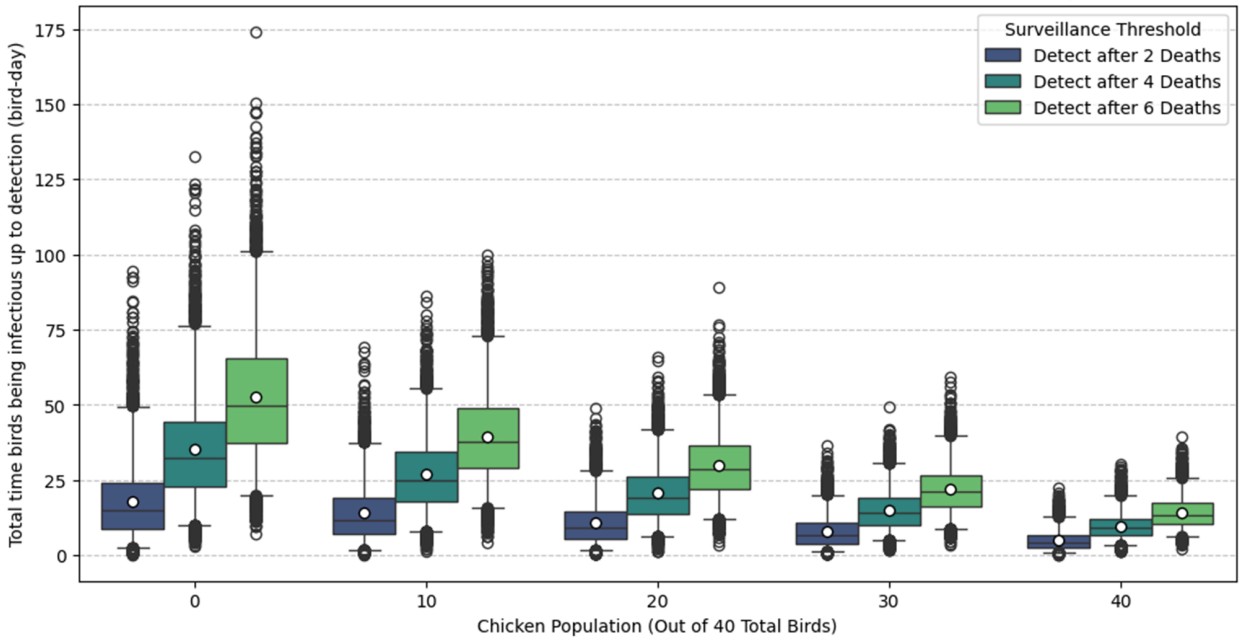

**Fig 7. Box plot burden of infection.** Undetected burden of infection (total time birds spent infectious up to the point of detection) for different death thresholds for detection and chicken population out of 40 total birds. Top plot: the duck's symptomatic probability $p_d = 0.2$; Bottom plot: the duck's symptomatic probability $p_d = 0.8$. The boxes indicate the interquartile range (25th and 75th percentiles), and the whiskers indicate the 2.5th and 97.5th percentiles. The white dot in the box indicates the average detection time, which also corresponds to the fourth and first row of Fig 8 (for detection after 2 deaths).

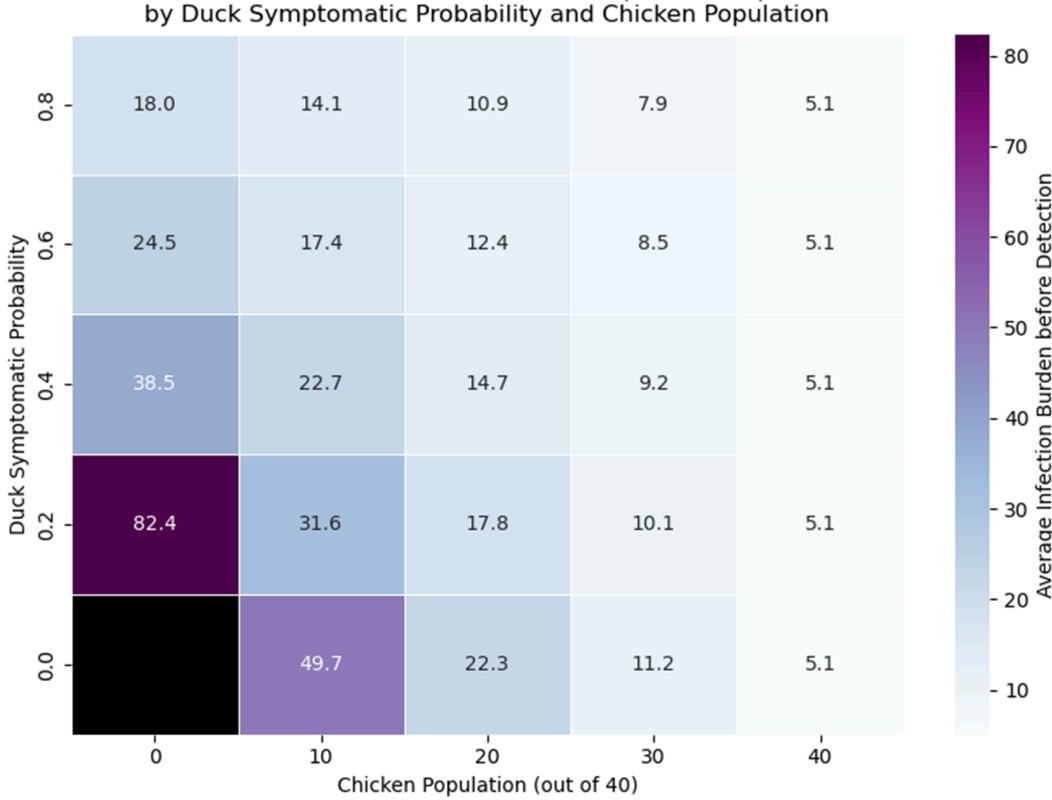

**Fig 8. Heat map burden of infection.** Heat map showing the average undetected burden of infections (with detection occurring when two deaths have occurred) in all simulation results given the parameter combinations of chicken populations and duck asymptomatic probability. Note that no detection would occur for duck flocks with duck symptomatic probability equal to zero (Black grid).

infection burden when we increased the value of $p_d$, which is potentially due to the reduced detection time. In the additional sensitivity analysis, we found that some parameters, such as $\beta_{dd}, \kappa$, and the total flock size, may have a moderate impact on the outcome upon detection. The choice of having 40 birds in our main simulations may also lead to an overestimation of detection time and burden of infection. However, we can also conclude that those parameters did not affect the result of our qualitative comparison between different configurations in a mixed-species flock.

One of our main assumptions regarding detection time is the criteria for detection. We assumed that a passive surveillance approach, where farmers voluntarily report a suspected infection in their flocks, has been implemented. We had chosen this approach as opposed to other strategies recommended for larger commercial flocks [26], because we believe that backyard farmers are more likely to have direct contact with the birds and therefore can observe signs and/or death reasonably efficiently. In particular, we believe that age-related mortalities are less likely to be identified as a sign of HPAI presence by backyard farmers. However, farmers' knowledge and attitudes towards HPAI can vary significantly and are difficult to quantify. A recent survey study by McClaughlin et al. [9] highlighted the lack of knowledge of the clinical signs of HPAI among UK birdkeepers. Another investigation by Elbers et al. shows that irregular mortality rate is one of the main criteria for voluntary reporting in the Netherlands [27]. Hence, we have chosen the detection criterion to be 'a certain number of deaths found in the flock'. We chose the threshold to be two, four, or six deaths to represent a likely range of farmer behaviour. These death thresholds are intended to reflect likely farmer behaviour in small backyard flocks, where voluntary reporting is common, whereas for commercial flocks it is often viable to have regulatory surveillance and laboratory

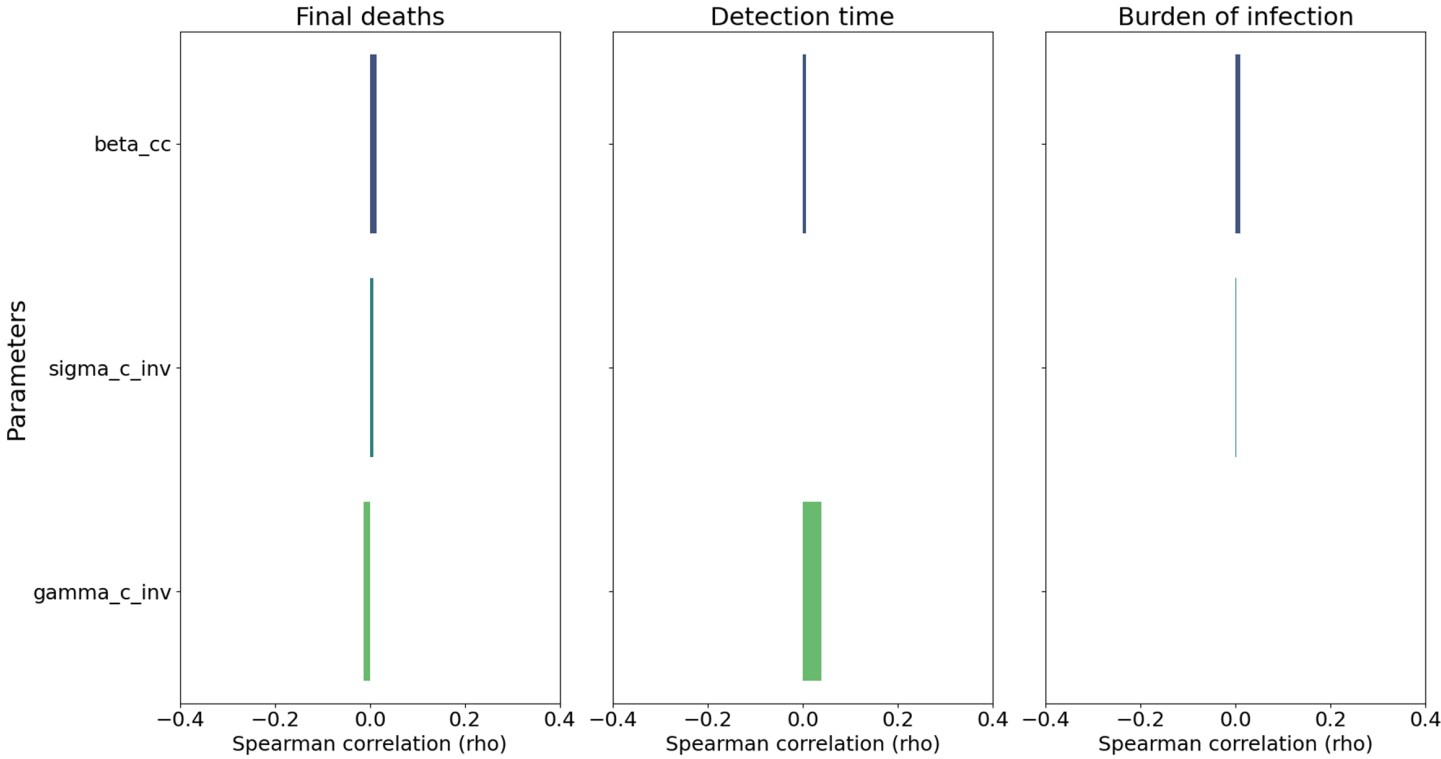

**Fig 9**. The Spearman correlation between the sampled parameters and final deaths (left), detection time (middle), and burden of infection (right), respectively, with fixed $p_d = 0.2$, and all 40 birds are chickens.

confirmation, which may detect infection at lower relative mortality or earlier clinical signs. However, a highly knowledge-able bird owner may be able to identify clinical signs in birds before their deaths, and a less knowledgeable farmer may only report when there is a large number of deaths. Some farmers may also be hesitant to report if the signs of HPAI is not sufficiently clear [28], or try to rapidly harvest and sell birds during an outbreak [29]. This may result in a larger vari-ance in detection time than what the simulation result suggests. Farmers' sensitivity to observed deaths can also lead to varied outcomes. Factors such as delayed reporting can delay the detection time, resulting in an underestimation of detection time and burden of infection from our results. To address the uncertainties of the detection criteria, more sur-veys and investigations are required to better quantify farmers' behaviours. For example, using surveys or interviews to group farmers by their knowledge level, reporting tendency, or response to flock mortality, as similarly demonstrated in Hill et al. [30].

Our choice of simulation model differs from some other studies on commercial farms [11,26] as we have excluded natu-ral deaths of birds in our model. Unlike in a commercial farm, where the process of raising birds can be highly automated, backyard hobby farmers tend to know and have close contact with their flocks. Hence, while natural and H5N1-induced mortality can be difficult to differentiate in a large-scale farm, we assumed that small-scale bird keepers can identify an 'unnatural' death with relative ease, so the existence of natural mortality in our model makes little impact. This assump-tion may slightly underestimate the total mortality rate of birds. However, background mortality rate is highly varied across different farm management styles and regions, making any assumed rates speculative and context dependent. We have added two plots in supporting information (S9 Fig) to show that for background mortality rate 0.001 days$^{-1}$ (approximately

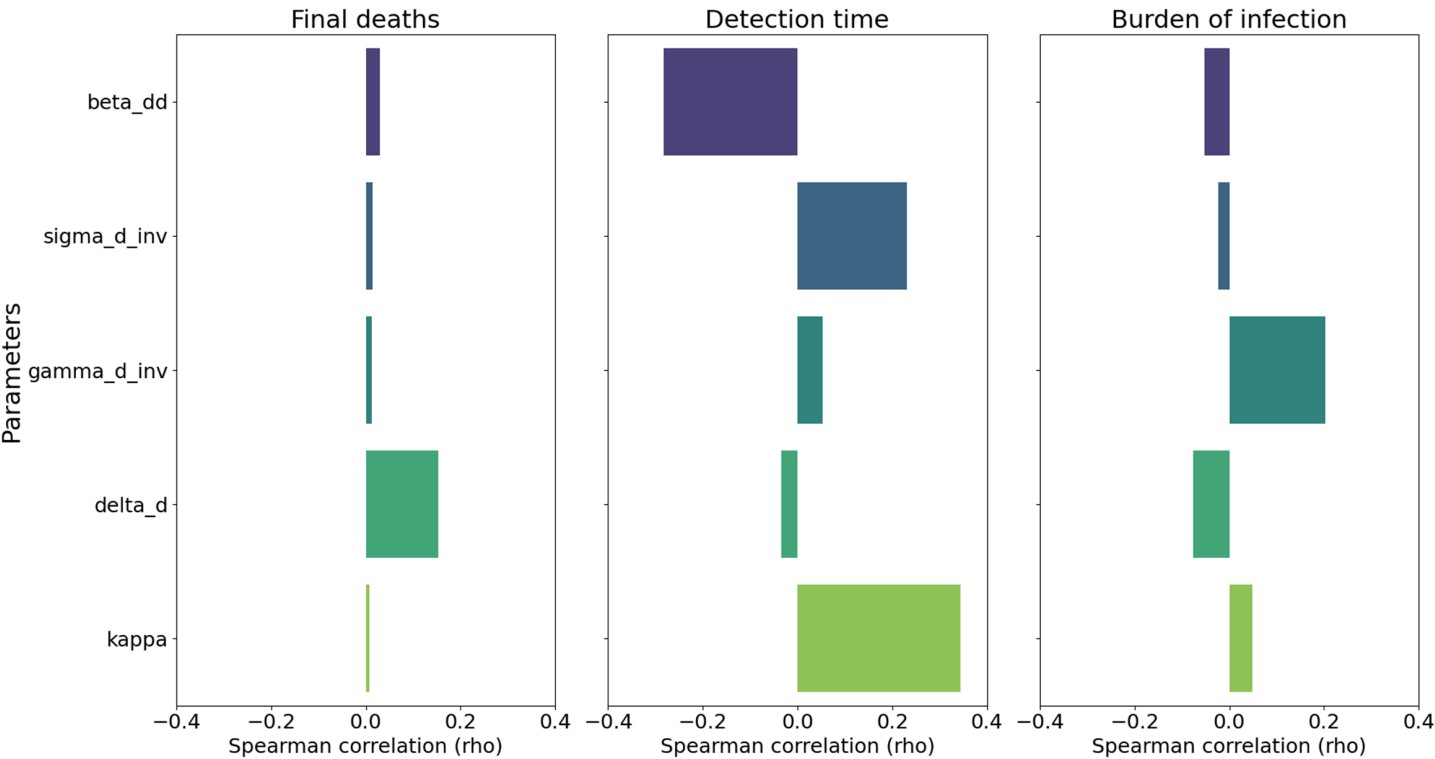

**Fig 10**. **The Spearman correlation between the sampled parameters and final deaths (left), detection time (middle), and burden of infection (right), respectively, with fixed** $p_d = 0.2$, **and all 40 birds are ducks.**

30% yearly mortality without HPAI), the impact of background mortality is minimal and does not change our qualitative results in any significant ways.

A limitation of our study is the choice of parameters in our model. Many parameter choices for our simulation were taken from either experimental studies (e.g., Bouma et al. [6]) or estimations from field data (e.g., Tiensin et al. [10], Vergne et al. [11]). Such parameters may change depending on the biosecurity measures and housing conditions of each individual flock. Additionally, the breeds and types of birds (e.g., broiler or layer) may also influence key parameters, which we did not account for in our simulations.

In our model, we assumed density-dependent transmission, such that the spread of disease through the flock depends on the total number of birds on the premises. In countries such as the UK, backyard poultry are typically kept in enclosures, and as flock size increases, the potential for contact also increases. However, in South East Asia, while some smallholders keep their poultry similarly enclosed, on other premises, birds are allowed to roam across much larger areas. In these cases, it may be more appropriate to assume frequency-dependent transmission, as birds are more likely to interact with a fixed number of contacts. The habitable space of the flock can also influence contact rates, so larger farms may have lower overall transmission rates than smaller farms. Furthermore, the inter-species transmission rate may be lower than assumed if chickens and ducks are kept separately, which violates our assumption of a homogeneously mixing population of birds. Some parameters, such as the probability of chickens showing clinical signs and dying from H5N1, were based on literature evidence and may vary in real-world settings. Additionally, our model had limited consideration of variance in farming conditions and environmental factors, which may impact its accuracy in specific backyard settings.

From a policy-making perspective, our findings indicate that species composition within backyard flocks has important implications for HPAI control strategies. Since passive surveillance leads to delayed detection and a higher risk of undetected transmission in duck-dominant flocks, enhanced measures such as active surveillance should be prioritised for these settings where the objective is to minimise the number of affected farms. In contrast, if reducing overall mortality is the primary goal, particularly in contexts where vaccination is available, prioritising interventions for chicken-dominant flocks is likely to be more effective due to their greater susceptibility and faster mortality. Species-specific strategies informed by our results can therefore help reduce both farm-level losses and between-flock transmission risk. We also recommend improving H5N1 awareness and symptom recognition among smallholders, particularly hobby farmers, to reduce the detection time and subsequently the burden of infections using the passive surveillance approach. Overall, these findings highlight the need for tailored surveillance and control policies that account for biological differences between bird species in backyard systems.

In conclusion, this study presents an exploratory analysis of how species composition may influence the within-flock transmission dynamics of HPAI. Our finding suggests that ducks, or waterfowl in general, are likely to play a more significant role in transmitting HPAI, while chickens or other similar species tend to have faster and more frequent mortalities, which is consistent with the current literature. Future work on parameter estimation of infection dynamics between various species of birds can be useful for further investigation. The parameter estimation may be integrated with real-time surveillance data and can be used on extended models such as spatial simulations or agent-based models. Another potential direction of research is to consider alternative surveillance and control strategies by taking the different transmissibility of H5N1 from various bird species into account. For example, a multi-objective optimisation problem comparing the trade-off between reducing mortality and limiting transmission can be investigated in the context of multi-species flocks, which would enable a more systematic analysis of different surveillance strategies.

## Supporting information

**S1 Fig. Infectious birds time series.** Number of infectious birds over time, for fixed $p_d = 0.2$. Red trajectories shows the time series directly, and the black trajectory is plotted by calculating the average across all time series on each day. Top: all 40 birds are ducks, middle: 20 chickens and 20 ducks, bottom: all 40 birds are chickens. (TIFF)

**S2 Fig. Dead birds time series.** Number of dead birds over time, for fixed $p_d = 0.2$. Blue trajectories shows the time series directly, and the black trajectory is plotted by calculating the average across all time series on each day. Top: all 40 birds are ducks, middle: 20 chickens and 20 ducks, bottom: all 40 birds are chickens. (TIFF)

**S3 Fig. Histogram final $p_d = 0.2$.** Histogram of distribution of final removed numbers (dead or recovered) with varied number of chickens in flocks of size 40. Duck symptomatic probability set to 0.2. The vertical dashed line indicates time series that are considered an outbreak (To the right of the dashed line). (TIFF)

**S4 Fig. Histogram final $p_d = 0.8$.** Histogram of distribution of final removed numbers (dead or recovered) with varied number of chickens in flocks of size 40. Duck symptomatic probability set to 0.8. The vertical dashed line indicates time series that are considered an outbreak (To the right of the dashed line). (TIFF)

**S5 Fig. Violin plot LHS all-chicken flock.** Violin plot for the distribution of final deaths, detection time, and burden of infections among all LHS simulations (all-chicken flocks). (TIFF)

**S6 Fig. Violin plot LHS all-duck flock.** Violin plot for the distribution of final deaths, detection time, and burden of infections among all LHS simulations (all-duck flocks).
(TIFF)

**S7 Fig. Bar plot outbreak/detection distribution.** Bar plot showing the probability of a simulation run resulting in an outbreak, and whether it would be detected with death thresholds 6, 4, or 2. This is fixed with $p_d = 0.2$. For $p_d = 0.8$ all outbreaks can be detected with the highest death threshold.
(TIFF)

**S8 Fig. Result vs total population size.** The mean detection time (top) and burden of infection (bottom) until detection for different total flock sizes and the proportion of chickens in the flock. Other parameters, including $p_d = 0.2$, were fixed.
(TIFF)

**S9 Fig. Simulation outcomes with natural mortality.** The detection time and burden of infection for the natural mortality rate $10^{-3}$ (approximately 30% yearly mortality without HPAI). This shows that the inclusion of natural mortality will not have any qualitative impact on our results.
(TIFF)

## Author contributions

**Conceptualization:** Steven Xingyu Wu.

**Data curation:** Steven Xingyu Wu.

**Formal analysis:** Steven Xingyu Wu.

**Methodology:** Steven Xingyu Wu, Christopher N. Davis, Michael J. Tildesley.

**Supervision:** Christopher N. Davis, Michael J. Tildesley.

**Validation:** Christopher N. Davis, Mark Arnold, Michael J. Tildesley.

**Writing – original draft:** Steven Xingyu Wu.

**Writing – review & editing:** Steven Xingyu Wu, Christopher N. Davis, Mark Arnold, Michael J. Tildesley.

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
