## [Decision Letter · Decision Letter 0]

22 Aug 2025

PCOMPBIOL-D-25-01462

The role of ducks in detecting Highly Pathogenic Avian Influenza in small-scale backyard poultry farms

PLOS Computational Biology

Dear Dr. Wu,

Thank you for submitting your manuscript to PLOS Computational Biology. After careful consideration, we feel that it has merit but does not fully meet PLOS Computational Biology's publication criteria as it currently stands. Therefore, we invite you to submit a revised version of the manuscript that addresses the points raised during the review process.

Please submit your revised manuscript within 60 days Oct 22 2025 11:59PM. If you will need more time than this to complete your revisions, please reply to this message or contact the journal office at ploscompbiol@plos.org. Please include the following items when submitting your revised manuscript:

We look forward to receiving your revised manuscript.

Kind regards,

Qiangguo Jin

Academic Editor

PLOS Computational Biology

Jennifer Flegg

Section Editor

PLOS Computational Biology

**Journal Requirements:**

At this stage, the following Authors/Authors require contributions: Steven Xingyu Wu, Christopher N Davis, Mark Arnold, and Michael J Tildesley. Please ensure that the full contributions of each author are acknowledged in the "Add/Edit/Remove Authors" section of our submission form.

3) Please ensure that the funders and grant numbers match between the Financial Disclosure field and the Funding Information tab in your submission form. Note that the funders must be provided in the same order in both places as well.

State what role the funders took in the study. If the funders had no role in your study, please state: "The funders had no role in study design, data collection and analysis, decision to publish, or preparation of the manuscript.".

**Reviewers' comments:**

Reviewer's Responses to Questions

**Comments to the Authors:**

Reviewer #1: In this manuscript, Wu et al. presents a well-designed simulation study examining the transmission dynamics of Highly Pathogenic Avian Influenza (HPAI) H5N1 within small-scale, mixed-species poultry flocks, with particular attention to backyard farming systems. The authors develop a computational model that simulates disease spread, incorporating factors such as species composition, chicken-to-duck ratios, mortality-based detection timing, and cumulative infectious burden prior to detection. This approach captures important complexities often overlooked in models designed for large, single-species commercial farms.

Overall, the manuscript is well written and addresses a timely and relevant topic. However, several aspects would benefit from further clarification and elaboration. Please see my comments below:

Major comments:

Abstract:

1. The authors could consider mentioning in the abstract that the authors have developed a computational model to simulate the spread of HPAI H5N1.

2. Including the GitHub link in the abstract would improve accessibility and encourage use of the model and data.

Introduction:

3. Add a brief (2–3 sentence) overview of why HPAI H5N1 matters globally and why backyard flocks are a unique concern.

4. Highlight novelty by emphasizing what makes your study different from existing research.

5. Briefly outline how the findings could inform public health or agricultural policy.

6. Discuss limitation by adding a short note on what the model does not capture (e.g., environmental factors, human intervention) adds credibility.

Methods:

7. The simulation-based approach is strong, with thoughtful use of the Gillespie algorithm and behavioral detection thresholds. However, the assumed flock size of 40 birds may limit generalizability. Absolute mortality thresholds (e.g., 2–6 birds) represent a large proportion of the flock and may not scale well to larger operations. Smaller flocks are also more susceptible to stochastic effects. To improve applicability, I suggest including simulations with larger flock sizes or a sensitivity analysis to assess how outcomes vary with population size.

8. The authors have acknowledged the key model assumptions such as homogeneous mixing and exclusion of natural mortality. It would be helpful to more clearly discuss how these assumptions might influence transmission dynamics and detection timing.

9. The authors have provided the GitHub link for data and code. To improve usability, they should consider adding a README file with clear instructions on how to run and interpret the model.

Results:

10. The differences between duck-only and chicken-only flocks are interesting. To strengthen the interpretation, the authors should more clearly explain how biological traits such as longer infectious periods or lower mortality in ducks relate to the simulation results.

11. The analysis of detection time at various death thresholds is valuable. To improve its practical relevance, it would help to discuss how these thresholds reflect real-world surveillance practices, particularly in smallholder and commercial settings.

12. Using bird-days to measure infectious burden is appropriate. To highlight its epidemiological importance, the authors could more clearly explain how a higher burden might increase the risk of transmission to nearby flocks or humans.

13. The decision to analyze only simulations with detected outbreaks may underestimate the true burden of infection, particularly in duck-only flocks. A brief discussion or sensitivity analysis of undetected outbreaks would provide a more complete picture.

Discussion:

14. The authors acknowledge variability in farmer behavior, which is commendable. However, the discussion could be strengthened by referencing more region-specific behavioral studies or proposing how behavioral heterogeneity could be modeled in future work.

15. The discussion mentions important assumptions, such as homogeneous mixing and the exclusion of natural mortality. However, their impact on the results should be explored in more depth. For instance, how could spatial separation or age-related mortality affect detection patterns?

16. The author could consider presenting the balance between reducing mortality and limiting transmission as a multi-objective optimization problem to strengthen the study’s computational relevance.

17. The proposed future work is appropriate but could be expanded to include agent-based models, spatial simulations, or integration with real-world surveillance data. Exploring these directions would broaden the study’s scope and enhance its impact.

Reviewer #2: The paper addresses an interesting research question related to modeling influenza transmission across different avian species. The manuscript is generally well written with a clear presentation of the methodology. However, after careful review, I have concerns regarding the significance of the problem, the novelty of the methodology, and the design of the numerical experiments.

Regarding the problem setup, the authors emphasize that interspecies transmission in avian populations has not been investigated. The focus of the article is on small backyard settings where there is limited or no data available to validate the parameter values used in the dynamical system models.

This raises concerns about the practical relevance and credibility of the simulation results. While small backyard scenarios are underexplored, higher-density industrial settings may be of greater interest from both scientific and public health perspectives.

Additionally, data from such settings may be available to support parameter estimation and model validation.

I think incorporating these would significantly enhance the attractiveness and trustworthiness of the study in real-world applications, or the authors should explain more clearly why they chose this setting given the limited data resources.

Regarding the methodology, the main novelty of the paper appears to lie in modeling transmission under a mixed-species structure. However, this novelty is weakened by the use of a simplified, uniform distribution of species.

Since this mixing structure is central to the contribution of the paper (putting aside the backyard setting), I suggest that the authors explore other mixing assumptions that reflect different contact probabilities or densities between species based on real-world scenarios. This would make the methodological contribution more convincing.

I do not see other significant methodological innovations that distinguish the manuscript, as the model is based on traditional compartmental formulations and simulations with parameter values drawn from previous references.

I think for a journal like PCB, some advances in computational methods should be proposed, or at least the authors should adopt other advanced existing computational frameworks to draw conclusions instead of relying solely on traditional methods.

For the numerical experiments, I have concerns about the validity of the results, as parameter values are taken from previous references that may not be fully suitable for backyard settings. Given that no data are available, the resulting conclusions have limited support from real-world scenarios.

I therefore suggest that the authors explore potential datasets to help validate the research findings, or change the problem settings to ones for which data validation is possible.

The design of the sensitivity analysis for some model parameters also confuses me, as only a few values are considered, which may be too simplistic for a highly nonlinear SEIR model with mixed densities assumed.

I suggest the authors switch to a continuous distribution for the sensitivity analysis instead of testing only a few points. Consider using Latin Hypercube Sampling if no data are available.

The purpose of sensitivity analysis should be to compare and identify parameters from the full set that are of interest. For example, people may be most interested in the transmission rate and asymptomatic probability, which are more sensitive and therefore more crucial for controlling the spread.

Based on this, a full comparison of sensitivities for all parameters is needed.

Reviewer #3: The review comments are uploaded as an attachment.

Reviewer #4: The paper by Wu et al. The role of ducks in detecting Highly Pathogenic Avian Influenza in small-scale backyard poultry farms, develops a SEIRD model and uses stochastic simulation to study avian influenza transmission between ducks and chickens in small mixed flocks. The parameters are drawn from literatures, and the simulations appear plausible. The work investigates the role of ducks in avian flu transmission in small population, which is interesting. And the authors have acknowledged the limitation of their work.

I have a few concerns:

1. In the text, beta^*_dc=beta_cc/kappa, however, it is inconsistent with kappa*beta_cc in Table 1. Please double check this and the corresponding simulation.

2. In detection time section, the work states that ‘due to higher mortality rate for ducks’, which is misleading as they specifically mentioned that chicken has nearly 100% death rate while the duck actually has a lower case-fatality rate. I think this statement requires clarification with biological evidence.

**Have the authors made all data and (if applicable) computational code underlying the findings in their manuscript fully available?**

Reviewer #1: Yes

Reviewer #2: Yes

Reviewer #3: Yes

Reviewer #4: Yes

PLOS authors have the option to publish the peer review history of their article (what does this mean?). If published, this will include your full peer review and any attached files.

Reviewer #1: **Yes:** Dr. Reema Singh

Reviewer #2: No

Reviewer #3: No

Reviewer #4: No

**Figure resubmission:**
---

## [Decision Letter · Decision Letter 1]

13 Nov 2025

PCOMPBIOL-D-25-01462R1

The role of ducks in detecting Highly Pathogenic Avian Influenza in small-scale backyard poultry farms

PLOS Computational Biology

Dear Dr. Wu,

Thank you for submitting your manuscript to PLOS Computational Biology. After careful consideration, we feel that it has merit but does not fully meet PLOS Computational Biology's publication criteria as it currently stands. Therefore, we invite you to submit a revised version of the manuscript that addresses the points raised during the review process.

We look forward to receiving your revised manuscript.

Kind regards,

Qiangguo Jin

Academic Editor

PLOS Computational Biology

Jennifer Flegg

Section Editor

PLOS Computational Biology

**Reviewers' comments:**

Reviewer's Responses to Questions

**Comments to the Authors:**

Reviewer #1: Thank you for addressing my comments. I appreciate the thoughtful revisions and the authors’ efforts to respond to the majority of the points raised. However, several comments appear to be only partially addressed and could benefit from further clarification or expansion to fully meet their intent.

1. Previous comment: Add a brief overview of why HPAI H5N1 matters globally and why backyard flocks are a unique concern.

New comment: The authors have partially addressed this by providing examples from the UK and Southeast Asia that highlight the elevated risk in backyard flocks. While this supports the concern about their vulnerability, it does not clearly explain the global significance of HPAI H5N1, such as its impact on public health, poultry industries, or zoonotic potential.

2. Previous comment: Highlight novelty by emphasizing what makes your study different from existing research.

New comment: Thank you for partially addressing this point. To more clearly highlight the novelty of your study, I recommend explicitly stating early in the discussion how your approach differs from prior work. For instance, you could emphasize the unique focus on backyard mixed-species flocks, your assumptions around passive surveillance, or specific modeling choices that distinguish your study.

3. Previous comment: Briefly outline how the findings could inform public health or agricultural policy.

New comment: Thank you for the additions that begin to connect your findings to policy relevance. To fully address the comment, I suggest summarizing how specific results, such as duck-related detection delays and species composition effects, could inform concrete public health or agricultural policies. A clear statement of practical implications would help readers understand how your findings can support decision-making.

4. Previous comment: The differences between duck-only and chicken-only flocks are interesting. To strengthen the interpretation, the authors should more clearly explain how biological traits such as longer infectious periods or lower mortality in ducks relate to the simulation results.

New comment: Thank you for adding a sentence to address this point. To strengthen the interpretation further, I suggest expanding the explanation of how these biological traits specifically influence simulation outcomes such as detection time, infection burden, and mortality patterns. This would help clarify the mechanisms driving the observed differences.

5. Previous comment: The analysis of detection time at various death thresholds is valuable. To improve its practical relevance, it would help to discuss how these thresholds reflect real-world surveillance practices, particularly in smallholder and commercial settings.

New comment: Thank you for partially addressing this comment. To improve practical relevance, I suggest briefly discussing how the selected death thresholds align with actual surveillance practices in both smallholder and commercial contexts. Referencing typical reporting criteria or known behavioral patterns could help contextualize the thresholds and strengthen the policy implications.

6. Previous comment: Using bird-days to measure infectious burden is appropriate. To highlight its epidemiological importance, the authors could more clearly explain how a higher burden might increase the risk of transmission to nearby flocks or humans.

New comment: Thank you for addressing this point. The added explanation strengthens the epidemiological relevance of bird-days. If applicable, consider briefly noting whether prolonged infectiousness in ducks could also elevate zoonotic risk to further clarify the public health implications.

7. Previous comment: The decision to analyze only simulations with detected outbreaks may underestimate the true burden of infection, particularly in duck-only flocks. A brief discussion or sensitivity analysis of undetected outbreaks would provide a more complete picture.

New comment: Thank you for including the bar plot in the supplementary materials. To fully address the comment, I suggest briefly discussing the implications of undetected outbreaks in the main text. For example, highlighting how these silent infections, especially in duck-only flocks, could influence surveillance strategies or underestimate transmission risk would reinforce the importance of your findings.

8. Previous comment: The authors acknowledge variability in farmer behavior, which is commendable. However, the discussion could be strengthened by referencing more region-specific behavioral studies or proposing how behavioral heterogeneity could be modeled in future work.

New comment: Thank you for citing relevant work on behavioral heterogeneity. To fully address the comment, I suggest referencing region-specific behavioral studies related to HPAI or smallholder poultry farming if available. Additionally, proposing how behavioral variability could be incorporated into future models, especially in the context of backyard flocks, would improve the relevance and depth of the discussion.

9. Previous comment: The discussion mentions important assumptions, such as homogeneous mixing and the exclusion of natural mortality. However, their impact on the results should be explored in more depth. For instance, how could spatial separation or age-related mortality affect detection patterns

New comment: Thank you for the update. To fully address the comment, I suggest briefly summarizing in the main text how key assumptions such as homogeneous mixing and exclusion of natural mortality might influence detection patterns. For example, spatial separation could slow transmission and delay detection, while age-related mortality might obscure outbreak signals. A clearer discussion of these mechanisms would help readers interpret the robustness of your findings.

10. Previous comment: The author could consider presenting the balance between reducing mortality and limiting transmission as a multi-objective optimization problem to strengthen the study’s computational relevance.

New comment: Thank you for acknowledging the suggestion. To fully address the comment, I recommend briefly elaborating on how the trade-off between reducing mortality and limiting transmission could be framed as a multi-objective optimization problem. Even a short mention of potential objectives, constraints, or modeling approaches would help clarify the computational relevance and open avenues for future work.

Reviewer #2: The manuscript has been substantially improved. I do not see any remaining issues and therefore accept the manuscript.

Reviewer #3: The authors have addressed most of my previous comments well. However, several of the responses require further discussion or additional clarification.

First, I am not fully convinced by the following response: “We believe that density-dependent models are more appropriate for small flocks. Unlike in large-scale farms, where each bird is more likely to have a fixed number of contacts regardless of population increase, in a small backyard the population size directly impacts the contact rate of birds.”

The extent of density-dependent transmission is influenced not only by the number of animals but also by the space available to them. While the combined effects of these factors are complex to understand, I believe that in backyard settings, birds are likely to have sufficient space to roam and express their social behaviours more freely. This could lead to more heterogenous mixing and frequency-dependent transmission, as birds would not need to interact with unrelated individuals against their social preferences. I acknowledge that when bird density is very high relative to the available space, movement may be restricted, and contacts may occur mostly among nearby individuals (so may have a fixed number of contacts regardless population increase as the authors wrote). However, such settings are unlikely to represent typical backyard settings I imagine. One could also favour the opposite dynamics (if density is high but birds can still move, they would have more contacts with related birds, resulting in density-dependent transmission).

Could you please provide a further justification for your assumptions regarding transmission mode and include that in the manuscript?

Second, please clarify why only layer chickens were chosen as the study population. Are layer chickens the most common type kept in backyard settings, and is this specific to the UK? I am not sure this assumption is generalisable, as in many Southeast Asian countries, for example, chickens are often raised primarily for meat production.

Finally, I remain unconvinced by the authors’ rationale for excluding background mortality. Background mortality is a key factor in passive surveillance on poultry farms, and detection time is one of the main outcomes of this study. If ducks and chickens have different background mortality rates, this could influence detection times to some extent. Exploring this aspect would strengthen the manuscript, or alternatively, please provide a more robust justification for not including it.

Reviewer #4: The authors have addressed my previous comments. I have no further comments.

**Have the authors made all data and (if applicable) computational code underlying the findings in their manuscript fully available?**

Reviewer #1: Yes

Reviewer #2: Yes

Reviewer #3: Yes

Reviewer #4: Yes

PLOS authors have the option to publish the peer review history of their article (what does this mean?). If published, this will include your full peer review and any attached files.

Reviewer #1: **Yes:** Dr. Reema Singh

Reviewer #2: No

Reviewer #3: No

Reviewer #4: No

**Figure resubmission:**
---

## [Editor Report · Decision Letter 2]

23 Dec 2025

Dear Mr Wu,

We are pleased to inform you that your manuscript 'The role of ducks in detecting Highly Pathogenic Avian Influenza in small-scale backyard poultry farms' has been provisionally accepted for publication in PLOS Computational Biology.

Best regards,

Qiangguo Jin

Academic Editor

PLOS Computational Biology

Jennifer Flegg

Section Editor

PLOS Computational Biology

---

## [Editor Report · Acceptance letter]

PCOMPBIOL-D-25-01462R2

The role of ducks in detecting Highly Pathogenic Avian Influenza in small-scale backyard poultry farms

Dear Dr Wu,

I am pleased to inform you that your manuscript has been formally accepted for publication in PLOS Computational Biology. Your manuscript is now with our production department and you will be notified of the publication date in due course.

With kind regards,

Anita Estes
